# Buffered 4% Articaine Reduces Pain and Enhances Anesthesia in Maxillary Third Molar Extractions: A Randomized, Double-Blind Split-Mouth Study

**DOI:** 10.3390/biomedicines12122691

**Published:** 2024-11-25

**Authors:** Henning Staedt, Victor Palarie, Diana Heimes, Peter Ottl, Shengchi Fan, Peer W. Kämmerer

**Affiliations:** 1Department of Prosthodontics and Materials Science, University Medical Center Rostock, Strempelstr. 13, 18057 Rostock, Germany; 2Private University in the Principality of Liechtenstein, Dorfstrasse 24, 9495 Triesen, Liechtenstein; 3Laboratory of Tissue Engineering and Cellular Cultures, Nicolae Testemitanu State University of Medicine and Pharmacy, 2004 Chisinau, Moldova; 4Department of Oral and Maxillofacial Surgery, Plastic Surgery, University Medical Center Mainz, Augtusplatz 2, 55131 Mainz, Germany; 5Oral Surgery and Implantology, Faculty of Medicine and Health Sciences, University of Barcelona, 08907 Barcelona, Spain

**Keywords:** articaine, buffers, local infiltration, third molars, tooth extraction

## Abstract

**Background/Objectives**: Buffered local anesthetics are suggested to enhance patient comfort and anesthetic efficacy during dental procedures. However, their advantages over non-buffered solutions in maxillary third molar extractions remain under-investigated. This prospective, randomized, double-blind, split-mouth study aimed to compare the efficacy of buffered versus non-buffered 4% articaine in terms of pain, anesthetic onset, and the duration of anesthesia during maxillary third molar extractions. **Methods**: Each participant randomly received two buccal infiltrations on two single appointments to extract maxillary third molars: one with 4% articaine buffered with sodium bicarbonate and the other with non-buffered 4% articaine. Pain during injection, post-extraction pain, onset time, and the duration of anesthesia were assessed using a visual analog scale (VAS; 0–10). **Results**: Fifty adult participants (23 males and 27 females) with a mean age of 21.6 years were included in the study. Buffered 4% articaine significantly reduced injection pain (mean VAS: 3.12 ± 1.36 vs. 4.2 ± 0.3, *p* = 0.0001) and post-extraction pain (mean VAS: 4.4 ± 1.75 vs. 5.76 ± 1.78, *p* = 0.0002) compared to non-buffered articaine. Additionally, buffered articaine demonstrated a faster onset of anesthesia (mean time (seconds): 85.92 ± 27.37 vs. 126.86 ± 33.15, *p* < 0.0001) and a longer duration of anesthesia (mean duration (minutes): 70.4 ± 13.64 vs. 51.4 ± 7.2, *p* < 0.0001). Regarding gender’s factor, the comparisons revealed no statistically significant differences (*p* < 0.05) in pain perception between males and females for either injection pain or post-extraction pain. **Conclusions**: Buffered 4% articaine provides superior anesthetic efficacy compared to non-buffered 4% articaine, enhancing patient comfort by reducing pain, accelerating onset, and prolonging the duration of anesthesia during maxillary third molar extractions.

## 1. Introduction

By reversibly blocking nerve conduction, local anesthetics (LAs) are essential for pain management in dentistry and enable a wide array of dental procedures [1]. Despite their essential role, the administration of local anesthetic injections often induces fear and anxiety among patients of all ages. This apprehension is particularly pronounced during tooth extractions, commonly associated with significant discomfort [2,3]. This anxiety often leads to the postponement of dental appointments and, in extreme cases, the refusal of necessary treatments. Dental phobia, particularly concerning wisdom tooth extractions, poses a major obstacle to patient compliance. To address these challenges, it is crucial to implement effective strategies that alleviate pain and anxiety while reducing the dependence on sedation [4,5]. Patients commonly report a burning or stinging sensation during local anesthetic injections, attributed to the acidic nature of commercially available anesthetics, which are typically adjusted to a pH of around 3.5 to 4.5 to stabilize the solution, particularly in formulations containing epinephrine [6]. Most anesthetic molecules in the cartridge exist in a water-soluble acidic state (RNH+), but optimal penetration of the nerve sheath requires the anesthetic to be in its unionized free base form (RN). This transition occurs through dissociating a hydrogen ion (H+) from the ionized molecule. It is hypothesized that reducing acid-sensing nociceptor activation by buffering the local anesthetic solution could lead to reduced injection pain [7]. Additionally, using alkalinized agents is believed to accelerate the transition from the ionized to the unionized form, promoting a more rapid onset of the anesthetic effect [8].

Numerous studies have shown that sodium bicarbonate-buffered lidocaine significantly reduces onset time and injection pain (VAS) compared to non-buffered lidocaine in inferior alveolar nerve blocks [9,10,11]. However, existing meta-analyses highlight statistical heterogeneity and small sample sizes, resulting in low to moderate evidence quality [12]. Recently, 4% articaine has gained widespread recognition for its superior diffusion through soft and hard tissues compared to other local anesthetics. Its efficacy and safety profile and reduced risk of systemic toxicity have led to its preferential use in dental procedures requiring profound anesthesia, such as molar extractions [13,14]. In contrast, research on buffered 4% articaine, widely regarded as the gold standard for dental local anesthesia in many countries, remains limited [12]. Shurtz et al. found no benefit in buffering 4% articaine with 8.4% sodium bicarbonate regarding anesthetic success, onset, or injection pain during primary buccal infiltration for mandibular first molars [15]. Dhake et al. reported reduced injection pain, faster onset, and less pain during extraction when buffered articaine was used for maxillary primary molar extraction in children [16]. Manta et al. also suggested that buffering 4% articaine with epinephrine could improve anesthetic properties, leading to faster onset and reduced injection pain [17]. However, no studies have specifically compared buffered and non-buffered 4% articaine in third molar extractions.

Therefore, this prospective, randomized, split-mouth study aimed to assess injection pain, the onset and duration of anesthesia, and pain during and after treatment, comparing 4% articaine with buffering against its non-buffered counterpart in patients requiring extractions of maxillary molars. The null hypothesis was that there is no significant difference between buffered and non-buffered 4% articaine regarding injection pain, anesthesia onset, duration, and post-treatment pain.

## 2. Materials and Methods

### 2.1. Patients

This prospective, randomized, split-mouth study was conducted in 2024 in the Department of Oral and Maxillofacial Surgery and Oral Implantology at Nicolae Testemitanu State University of Medicine and Pharmacy, Moldova (ethical approval number: 41/03.02.2020). The study received approval from the Ethics Committee of Chisinau, Moldova. Patients were selected based on the following inclusion criteria:Participants aged 18 to 88 requiring the extraction of maxillary third molars on both sides (teeth 18 and 28), classified as A1 according to Pell’s and Gregory’s classification (fully erupted molars [18]);no presence of acute inflammation or symptomatic pulpitis in the maxillary third molars;good oral and general health (ASA-classification I-II) without contraindications for dental extractions;no known allergies or adverse reactions to articaine or related anesthetics;absence of significant cardiovascular, hepatic, renal, or respiratory diseases; andno history of medical or psychological conditions that may hinder compliance with study requirements.

The exclusion criteria were as follows:Presence of contraindications for dental extractions, such as uncontrolled bleeding disorders;requirement for maxillary molar surgery involving surgical access and osteotomy;history of allergies or adverse reactions to articaine or related local anesthetics;significant cardiovascular, hepatic, renal, or respiratory diseases;pregnancy or breastfeeding;regular use of medications that may interfere with pain perception or anesthesia response, unless on a stable dose for at least 30 days (specific medications of concern were analgesics and anticonvulsants); andcommunication barriers or language limitations that may hinder effective participation (assessed through a pre-study interview).

Enrolled patients provided written informed consent to participate, and appointments were scheduled for examination and surgery. Pain levels for each maxillary third molar were assessed using a visual analog scale (VAS) immediately after the anesthesia injection and 3 h post-extraction. Patients self-reported the pain levels through a questionnaire administered by a clinician. The classification of maxillary third molars was determined and documented through clinical and radiographic examinations following the Pell and Gregory classification system [18].

### 2.2. Study Design

In this prospective, randomized, double-blind study, all participants received two injections consisting of a single primary buccal infiltration for the maxillary third molar using either 2 mL of 4% articaine with 1:200,000 epinephrine (Sopira^®^ Citocartin, Kulzer, Hanau, Germany) or 1.8 mL of 4% articaine mixed with 0.2 mL of 8.4% sodium bicarbonate (NaBic) (B. Braun, Melsungen, Germany). These injections were administered during two single and separate appointments for each side of the maxilla, with a minimum interval of two weeks between procedures. Randomization was performed using a computer-generated random sequence, assigning each extraction to either the buffered 4% articaine group or the non-buffered 4% articaine group. To maintain the double-blind approach, the anesthetic solutions were prepared by an independent dental nurse who was not involved in the clinical procedures. The syringes containing the anesthetic solutions were identical in appearance and were labeled only with a code corresponding to the randomization sequence, ensuring that neither the clinician nor the patient was aware of which solution was administered.

### 2.3. Preparation and Extraction

In the buffered group, under sterile conditions, 0.2 mL from a 1.8-mL cartridge of 4% articaine (with 1:200,000 epinephrine) was removed and replaced with 0.2 mL of 8.4% sodium bicarbonate (NaBic) using an insulin syringe [19], following a protocol to ensure proper mixing of the solutions. Before each injection, the mucosa was dried with gauze, and 0.2 mL of topical anesthetic spray (Xylocain^®^, Aspen, Munich, Germany) was applied with a cotton tip applicator for 60 s at the injection site. The injection target site was centered over the buccal root apices of each maxillary third molar, and the anesthetic solution was deposited slowly over 1 min. All extractions were performed using elevators and forceps. Patients requiring osteotomy during the procedure were excluded from the study. All injections and extractions were performed by the same surgeon (SH).

### 2.4. Outcomes Evaluation

#### 2.4.1. Primary Outcome Parameter

The primary outcome measured was pain during injection, assessed using a visual analog scale (VAS). Participants were asked to rate their pain on an 11-point VAS, where 0 represented no pain, and 10 represented the maximum pain immediately after needle withdrawal. Participants were instructed on how to use the VAS before the procedure to ensure consistent reporting.

#### 2.4.2. Secondary Outcome Parameters

Post-extraction pain was measured using a VAS 3 h after the extraction [10]. Patients were advised not to take painkillers within this period to avoid bias in reporting pain levels. Additionally, the onset of anesthesia was measured by recording the time taken for the anesthetic to take effect, using an ice spray applied to the buccal mucosa every 5 s, starting 30 s after injection, until the patient no longer felt the probe. Tooth extraction was only performed once the patient reported no pain. The duration of anesthesia was recorded during the subsequent visit, based on the patient’s report of the time when pain resumed or when they required a rescue analgesic. Patients were instructed to keep a pain diary to document when they first experienced pain or took medication.

### 2.5. Statistical Analysis

The difference in the proportion of injection pain was considered the primary outcome for determining the effect size. Based on the results of Dhake [16], an effect size of approximately 2.5 with a standard deviation of about 2 was anticipated. The sample size was estimated using the following assumptions: an alpha error of 5% and a study power (1-beta) of 80%, with Z-scores of approximately 1.96 for alpha/2 and 0.84 for beta in a two-tailed test. This calculation yielded a sample size of 10 participants. Given that the study employed a split-mouth design, where each participant served as their control, the total number of participants required would be doubled, resulting in at least 20 participants per group. The data was analyzed using *t*-tests and the Mann–Whitney U test to determine statistical significance. IBM SPSS Statistics for Windows, Version 22.0 (Released 2013; IBM Corp; Armonk, NY, USA) and GraphPad Prism 6.0, released in August 2012 (Dotmatics, Boston, MA, USA), were used for the statistical analysis. The presence or absence of a *p*-value of 0.05 determined the level of statistical significance.

## 3. Results

Fifty-five adult participants were initially enrolled in the study. Five participants were excluded because a surgical approach became necessary during the procedure. Thus, 50 adults (23 men and 27 women) aged 18–42 years, with a mean age of 21.6 years, completed the study. A total of 100 infiltrations and extractions were performed. Following the randomization protocol, 50 infiltrations of 4% articaine were administered, with 29 on the right third molar and 21 on the left third molar.

In comparison, 50 buffered articaine infiltrations were administered, with 21 on the right and 29 on the left. No adverse events or complications related to the anesthetic injections or extractions were reported during the study. Descriptive statistics for participant characteristics are provided in Table 1. After the study, both participants and clinicians were asked to guess whether buffered or non-buffered solutions had been used to assess the effectiveness of the blinding. Neither group could reliably distinguish between the solutions, confirming the success of the blinding protocol.

The analysis of subjective pain assessments for injections and post-extraction pain, evaluated using the VAS for maxillary third molar infiltrations, showed that pain scores were significantly lower in the buffered group compared to the control group (Table 2). The assessment of anesthesia onset demonstrated that it was significantly faster in the buffered group compared to the control group (Table 2). Additionally, the duration of anesthesia was significantly longer on the sides where buffered solutions were administered (Table 2).

Most of the data, particularly for injection pain, post-extraction pain, and onset time, did not follow a normal distribution according to the Shapiro–Wilk test. While the data for anesthesia duration appeared normally distributed, the Mann–Whitney U test was applied to all parameters for consistency. The results showed significant differences in the onset and duration of anesthesia, with the buffered solution demonstrating a faster onset and a longer duration. Additionally, pain scores, including both injection pain and post-extraction pain, were significantly lower in the buffered group (Table 2).

Regarding gender factors (Table 3), both male and female participants reported significantly lower (*p* < 0.05) injection and post-extraction pain levels in the buffered group compared to the control group. The assessment of anesthesia onset showed that it was significantly faster (*p* < 0.05) in the buffered group compared to the control group in both male and female groups (Table 3). The duration of anesthesia was significantly longer (*p* < 0.05) on the sides where buffered solutions were administered in both male and female groups (Table 3). There were no statistically significant differences in injection pain and post-extraction pain between male and female participants within either the buffered or control groups for any treatment. However, there were significant differences (*p* = 0.028) in the duration of anesthesia between male and female participants in the buffered group. The difference in the non-buffered group anesthesia duration between males and females was not statistically significant, with a *p*-value of 0.054. Additionally, the differences in onset time between males and females in the buffered and control groups were not statistically significant, with *p*-values of 0.054 and 0.711, respectively.

## 4. Discussion

Achieving effective anesthesia before dental treatment is critical for ensuring patient cooperation and reducing anxiety during the procedure [20]. The discomfort experienced by the patient depends on both their individual pain threshold and the effectiveness of the local anesthetic [21]. Buffering local anesthetics is based on the Henderson–Hasselbalch equation, which explains that adjusting a solution’s pH closer to its pKa increases the availability of the free base form, improving its ability to penetrate the nerve sheath. This randomized controlled trial involving 50 adult participants demonstrated that buffered articaine significantly improves patient comfort, speeds up the onset of anesthesia, and prolongs its duration, thereby enhancing the overall efficacy of anesthesia for maxillary third molar infiltrations.

Few studies have used the same anesthetic formula in different clinical settings. In Dhake’s study, 70 children aged 4–10 years who required local anesthesia for primary maxillary molar extractions were randomly assigned to either the buffered articaine group or the non-buffered group [16]. The study concluded that buffered articaine provided superior comfort, faster anesthetic onset, and better pain management in pediatric patients. However, the parallel-arm design might have introduced variability due to inter-individual differences, particularly considering the heightened discomfort or anxiety that children may experience when undergoing multiple dental procedures. In contrast, the split-mouth design of our study allowed each participant to serve as their own control, which effectively minimized variability associated with individual differences in pain thresholds. In Shurtz’s study, which employed a crossover design, 80 adult subjects each received two buccal infiltrations (one buffered and one non-buffered) for primary buccal infiltration in the mandibular first molar [15]. The study concluded that buffering 4% articaine did not provide any significant advantage over the non-buffered solution regarding anesthetic success, onset of anesthesia, or injection pain. The authors suggested that the body’s intrinsic buffering system rapidly neutralizes the pH of injected anesthetic solutions, which could explain why the buffered formulation did not demonstrate significant improvements [22,23]. In contrast, the benefits observed in the present study might be attributed to anatomical differences between the maxilla and mandible, where the more porous maxillary bone may allow for faster diffusion of the buffered solution. However, this may not fully explain the lack of significant difference in immediate injection pain between the two formulations observed in their study. In contrast, the present study found that buffered 4% articaine demonstrated positive outcomes across all evaluated parameters, including faster onset, reduced pain, and longer duration of anesthesia. Future research should further explore these findings, potentially examining different clinical scenarios better to understand the pharmacology and potential benefits of buffered anesthetics.

More studies have focused on buffered anesthetics in nerve block injections. A meta-analysis of 11 studies concluded that buffered lidocaine significantly reduces both injection pain and onset time during inferior alveolar nerve block injections compared to non-buffered lidocaine, though the evidence quality is considered moderate, mainly due to small sample sizes and study heterogeneity, highlighting the need for further high-quality research [12]. Bala et al.’s study further supported these conclusions, emphasizing the clinical relevance of faster onset and significantly lower injection pain, with pH adjustment being a key factor [9]. This process results in a faster onset of action and reduces injection pain, as the solution is less acidic and, therefore, less irritating to the tissues. In addition to confirming reductions in pain levels and onset time, Jain et al.’s study uniquely reported a significant increase in the duration of anesthesia with buffered formulations [10].

The significance of faster onset of local anesthesia in clinical practice has been widely documented in the literature [24,25,26]. Studies suggest that buffered anesthetic agents’ quicker onset of action may also contribute to reduced pain during injection. However, several practical challenges associated with buffered anesthetics in clinical settings remain. The preparation process requires additional time and technical expertise, which can be demanding in a busy clinical environment. Additionally, buffered solutions have a shorter shelf life and decreased stability compared to non-buffered anesthetics, necessitating prompt use after preparation [26]. Bala et al. also noted that the pH of buffered solutions was not measured after mixing, potentially introducing variability in effectiveness if the buffering process is not performed precisely [9]. Although there is a theoretical concern that incorrect buffering could reduce anesthetic potency, no study has observed this so far. While few pre-mixed buffered anesthetic products are available, such as Onset^®^ (Onpharma; Los Gatos, CA, USA), these products aim to simplify the buffering process by providing ready-to-use formulations, which may address some of these practical challenges in clinical practice.

Some evidence suggests that pain perception varies across different gender and age groups. In Shurtz’s study, pain levels were statistically higher in females than in males [15]. However, the present study did not find a statistically significant difference in pain perception between genders, a finding consistent with Bala’s study [9]. Regarding age, while one study suggested that older patients may be more sensitive to intense stimuli than younger individuals, other studies have reported no significant differences in pain perception across age groups. Arora’s study similarly found no significant age-related differences in pain perception, suggesting that age is unlikely to have influenced between-group comparisons when evaluating injection and procedure-related pain [27]. In the present study, although the goal was to include adults aged 18 to 88, the sample predominantly consisted of younger participants, with a mean age of 21. Consequently, it was impossible to assess the relationship between pain perception and age groups, which represents a limitation of this study.

Despite the limitations related to age and gender distributions, the study concluded that buffered 4% articaine demonstrated advantages over non-buffered 4% articaine in terms of reducing injection pain, post-extraction pain, and enhancing anesthesia onset and duration for primary buccal infiltration of the maxillary third molar. However, future studies must explore unanswered questions, such as whether buffered articaine can improve injection pain relief and onset time in patients with acute inflammation. Additionally, more randomized controlled trials with larger and more diverse populations are necessary to validate these findings further.

## 5. Conclusions

The results of this study confirmed that using buffered 4% articaine for infiltration of the maxillary third molar can enhance anesthesia efficacy and improve patient comfort by reducing injection pain and post-extraction pain, accelerating the onset of anesthesia, and prolonging its duration. Sodium bicarbonate as an adjunct in 4% articaine is an effective, safe, and clinically accessible solution. However, further studies are recommended to assess its benefits in different clinical scenarios, such as cases involving acute inflammation or more diverse patient populations, to fully validate its efficacy across a broader range of conditions.

## Figures and Tables

**Table 1 biomedicines-12-02691-t001:** Descriptive statistics for participant characteristics, including gender distribution, age range, and the number of infiltrations.

Characteristics	Numbers	Distribution
Total participants	50	
Gender	23 men 27 women	46% 54%
Mean age (years)	21.6	/
Age range (years)	18–42	/
Total infiltrations	100	
4% articaine	50	50%
18	29	29%
28	21	21%
Buffered 4% articaine	50	50%
18	21	21%
28	29	29%

**Table 2 biomedicines-12-02691-t002:** Comparison of pain at the injection sites, post-extraction pain, the onset of action, and the duration of anesthesia between buffered 4% articaine and 4% articaine.

Outcomes	4% Articaine (n = 50)	Buffered 4% Articaine (n = 50)	Mean Difference	*p* Value	Mann–Whitney U test	Z Value
Injection pain (mean VAS score)	4.2 ± 1.31	3.12 ± 1.36	−1.08	0.000124	1794.0	3.75
Post-extraction pain (mean VAS score)	5.76 ± 1.78	4.4 ± 1.75	−1.36	0.000629	1740	3.38
Onset time (in seconds)	126.86 ± 33.15	85.92 ± 27.37	−40.94	<0.0001	2042.5	5.46
Duration of anesthesia (in minutes)	51.4 ± 7.20	70.4 ± 13.64	19	<0.0001	252.5	−6.88

**Table 3 biomedicines-12-02691-t003:** Comparison between males and females in the group of buffered and control groups.

Outcomes	4% Articaine (n = 50)		Buffered 4% Articaine (n = 50)	
	Male (n = 23)	Female (n = 27)	Male (n = 23)	Female (n = 27)
Injection pain (mean VAS score)	3.96 ± 1.33	4.41 ± 1.28	2.91 ± 1.44	3.30 ± 1.30
Post-extraction pain (mean VAS score)	5.35 ± 1.82	6.11 ± 1.69	4.09 ± 1.88	4.67 ± 1.62
Onset time (in seconds)	124.91 ± 37.31	128.52 ± 29.78	78.09 ± 21.14	92.59 ± 30.55
Duration of anesthesia (in minutes)	49.26 ± 7.34 ^#^	53.22 ± 6.68 ^#^	65.96 ± 10.36 *	74.19 ± 15.08 *

* *p* < 0.05, ^#^
*p* = 0.054.

## Data Availability

The original contributions presented in the study are included in the article, further inquiries can be directed to the corresponding authors.

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
