# Peer review of "Buffered 4% Articaine Reduces Pain and Enhances Anesthesia in Maxillary Third Molar Extractions: A Randomized, Double-Blind Split-Mouth Study"

_biomedicines, 2024, doi:10.3390/biomedicines12122691_

Round 1

Reviewer 1 Report

Comments and Suggestions for Authors

The article entitled"  Buffered 4% Articaine Reduces Pain and Enhances Anesthesia in Maxillary Third Molar Extractions: A Randomized, Double- Blind Split-Mouth Study" was well written.

some minor concerns:

1) have you registered this study in clinical trials?

2) do you have any Ethical approval number by an ethics committee?

3) Was pain measurement after 3 hours verbal or they wrote or mark the number?

4) As level of pain threshold is different for each participant, how do you compare the results without having a history of pain tolerance of each?

Author Response

Reviewer 1

Comment #1: The article entitled "Buffered 4% Articaine Reduces Pain and Enhances Anesthesia in Maxillary Third Molar Extractions: A Randomized, Double-Blind Split Mouth Study" was well written.

Our response: Thank you for your review and positive feedback on our manuscript.

Comment #2: Have you registered this study in clinical trials?”

Our response: This study was not registered in clinical trials. We did not think of it and a retrospective registration does not make sense. However, we did not see that this is mandatory in this journal.

Comment #3: Do you have any Ethical approval number by an ethics committee?”

Our response: We submitted the ethical approval letter separately to the Editorial Office of the Journal. The respective approval number was added to the manuscript.

Comment #4: Was pain measurement after 3 hours verbal or they wrote or mark the number?”

Our response: The participants wrote the number with visual analog scale. This is explained in the manuscript more clearly now.

Comment #5: As level of pain threshold is different for each participant, how do you compare the results without having a history of pain tolerance of each?”

Our response: We appreciate this insightful question. The split-mouth design of our study allowed each participant to serve as their own control, which effectively minimized variability associated with individual differences in pain threshold. This was also added to the discussion section.

Reviewer 2 Report

Comments and Suggestions for Authors

Dear authors,
Thank you for giving me a chance to review your manuscript entitled " Buffered 4% Articaine Reduces Pain and Enhances Anesthesia in Maxillary Third Molar Extractions: A Randomized, Double-Blind Split-Mouth Study".

Kindly find my comments below which may help you.

Introduction

The background of this study is well described and the objective is clear.

P2L90: A period is missing.

Materials and Methods

1. What is the order of extraction of the left and right teeth? Did the authors perform both left and right extractions at the same time in one day? What were the reasons for this? The anesthetic effects and pain of one side may affect the other side.

2. The authors reported that random sequencing was used to assign either the buffered 4% articaine group (BA) or the non-buffered 4% articaine group (NBA). However, the results show that each was performed 50 times. Wasn't it the left or right site that was randomized, not the type of anesthetic? If randomization were based on the type of drug, there would be four possible combinations (BA: BA, BA: NBA, NBA: BA, NBA: NBA). Is this result 29:21? The authors should provide clear explanations to the readers.

Results

1. P5L203, P9L280: A period is missing.

2. The authors should remove Figures 1, 2, 5, and 6, as their results are overlapped in Table 2.

3. The authors should summarize Figures 3 and 4 in a table, including the results for onset time and duration.

4. Table 2 and Table 3 should be unified as they overlap.

Author Response

Reviewer 2

Comment #1: Introduction. P2L90: A period is missing.”

Our response: Correction done.

Comment #2: What is the order of extraction of the left and right teeth? Did the authors perform both left and right extractions at the same time in one day? What were the reasons for this? The anesthetic effects and pain of one side may affect the other side.“

Our response: The extractions on the left and right sides were conducted during separate appointments, with a minimum interval of two weeks between procedures. This approach was chosen to ensure adequate recovery time for each side and to minimize potential crossover effects, such as anesthetic influence and residual pain from one side impacting the other. By allowing a sufficient interval between procedures, we aimed to reduce any confounding effects and ensure more accurate assessments of pain and healing on each side independently. This is written in the manuscript more clearly, now.

Comment #3: „The authors reported that random sequencing was used to assign either the buffered 4% articaine group or the non-buffered 4% articaine group. However, the results show that each was performed 50 times. Wasn't it the left or right site that was randomized, not the type of anesthetic? If randomization were based on the type of drug, there would be four possible combinations (BA: BA, BA: NBA, NBA: BA, NBA: NBA). Is this result 29:21? The authors should provide clear explanations to the readers.”

Our response: Thank you for highlighting this point. In our study, the left or right side was randomized for each participant, and the assigned side then received either the buffered (BA) or non-buffered (NBA) 4% articaine accordingly. This approach ensured that each participant received both types of anesthetic on different sides, maintaining internal control within each individual. Consequently, each treatment type (BA and NBA) was administered 50 times across the cohort. This method allowed us to control for side-based variability while comparing the effects of the buffered and non-buffered anesthetics within the same participants.

Comment #4: Results. P5L203, P9L280: A period is missing.”

Our response:  We apologize. A period has been added to ensure grammatical accuracy.

Comment #5: Results. The authors should remove Figures 1, 2, 5, and 6, as their results are overlapped in Table 2.”

Our response: We agree to streamline the presentation and remove the specified figures to avoid redundancy, focusing on Table 2 for a comprehensive data view.

Comment #6: Results. The authors should summarize Figures 3 and 4 in a table, including the results for onset time and duration.”

Our response: We have created a supplementary table 3 that includes the summarized data on onset time and duration for each group, providing an easier reference for readers to compare the results directly.”

Comment #7: Table 2 and Table 3 should be unified as they overlap.”

Our response: We have unified the tables to present the overlapping data in a single, comprehensive table 2, allowing for a more streamlined presentation of the results

Round 2

Reviewer 2 Report

Comments and Suggestions for Authors

Dear authors,

Thank you for giving me a chance to review your manuscript entitled " Buffered 4% Articaine Reduces Pain and Enhances Anesthesia in Maxillary Third Molar Extractions: A Randomized, Double-Blind Split-Mouth Study".

All the issues that I previously pointed out are now revised. Manuscript has improved. I appreciate author hard work.